# Aetiological and prognostic roles of frailty, multimorbidity and socioeconomic characteristics in the development of SARS-CoV-2 health outcomes: protocol for systematic reviews of population-based studies

Tatjana T Makovski [1], Jinane Ghattas,[2] Stephanie Monnier Besnard,[1] Monika Ambrozova,[3] Barbora Vasinova,[3] Rodrigo Feteira-Santos,[4,5] Peter Bezzegh,[6] Felipe Ponce Bollmann,[7] James Cottam,[8,9] Romana Haneef,[1] Brecht Devleesschauwer,[8,10] Niko Speybroeck,[2] Paulo Nogueira [4,5] Maria João Forjaz,[11] Joel Coste,[1] Laure Carcaillon-Bentata[1]

**Correspondence to**
Dr Tatjana T Makovski; tatjana.makovski@ santepubliquefrance.fr

## ABSTRACT

**Introduction** There is growing evidence that the impact of COVID-19 crisis may be stronger for individuals with multimorbidity, frailty and lower socioeconomic status. Existing reviews focus on few, mainly short-term effects of COVID-19 illness and patients with single chronic disease. Information is also largely missing for population representative samples.

Applying population-based approach, the systematic reviews will have two objectives: (1) to evaluate the aetiological roles of frailty, multimorbidity and socioeconomic status on SARS-CoV-2 infection probability, hospitalisation, intensive care unit (ICU) admission, mechanical ventilation and COVID-19 related mortality among general population and (2) to investigate the prognostic roles of frailty, multimorbidity and socioeconomic characteristics on the risk of hospitalisation, ICU admission, mechanical ventilation, COVID-19 mortality, functioning, quality of life, disability, mental health and work absence.

**Methods and analysis** For this ongoing work, four databases were searched: PubMed, Embase, WHO COVID-19 Global literature on coronavirus disease and PsycINFO, for the period between January 2020 and April 7 2021. Peer-reviewed published literature in English and all types of population-based studies will be considered. Studies using standard tools to assess multimorbidity such as disease count, comorbidity indices or disease combinations will be retained, as well as studies with standard scales and scores for frailty or measurement of a socioeconomic gradient. Initial search included 10 139 articles, 411 for full-text reading. Results will be summarised by risk factor, objective and outcome. The feasibility of meta-analysis will be determined by the findings and will aim to better understand uncertainties of the results. Quality of studies will be assessed using standardised scales.

## STRENGTHS AND LIMITATIONS OF THIS STUDY

⇒ This Preferred Reporting Items for Systematic Reviews and Meta-Analyses guided protocol describes methodology of systematic reviews regarding the roles of multimorbidity, frailty and socioeconomic risk factors in development of SARS-CoV-2 health outcomes.

⇒ Only quantitative studies conducted on population representative samples will be eligible.

⇒ Only peer-reviewed literature published in English will be considered.

⇒ The literature screening process, data extraction and study quality assessment will be performed in pairs.

⇒ The reviews may be limited with the heterogeneity of available studies in terms of diversity of measurement tools used to assess risk factors and outcomes.

**Ethics and dissemination** The study will be based on published evidence, and it is exempt from the ethical approval. This work is part of the Population Health Information Research Infrastructure (PHIRI) project. Dissemination of the results will imply conference presentation, submission for scientific publication and PHIRI project report.

**PROSPERO registration number** CRD42021249444.

## INTRODUCTION

The early months of 2020 were marked with rapid growth in infections caused by the novel SARS-CoV-2, leading to the state of a global pandemic declared by the WHO on 11 March 2020.[1] SARS-CoV-2 causes COVID-19,[2]

which has led to severe health and healthcare management consequences around the world.[3]

It has been demonstrated so far that age, male sex and comorbidity are the main risk factors for COVID-19 hospitalisations, intensive care unit (ICU) admissions and mortality.[4–6] Specifically, infection fatality rates due to COVID-19 have been found to exponentially increase with age : 0.4% at age 55 years, 1.4% at age 65 years, 4.6% at age 75 years and 15% at age 85 years.[7] In addition, having at least one comorbidity more than doubled the risk of hospitalisation, while certain comorbidities, including severe kidney disease, diabetes, severe immunodeficiency or heart failure, among few others, increased this risk in particular, for example, OR/HR ≥3 for hospitalisation or death.[8] Much remains to be clarified due to the lifespan of the new disease of only 2 years and the appearance of new virus variants[9] against which the same protection level conferred by existing vaccines is not guaranteed;[10] for example, which and why certain population groups still display more severe symptoms and outcomes compared to others.

Frailty, as a marker of increased vulnerability due to accumulated deficits in multiple body systems,[11] and multimorbidity, commonly defined as having two or more chronic conditions,[12] are well recognised risk factors for adverse health outcomes in the elderly population, including hospitalisation, dependency and mortality.[13–15] In the context of the COVID-19 pandemic, it is plausible that both frailty and multimorbidity increase the risk of SARS-CoV-2 infection and the development of severe outcomes by altering older people's biological background and immunity. Some studies have found that frailty and multimorbidity predicted COVID-19 patients' prognosis for hospitalisation, mortality and level of care at discharge.[16–26] To date, however, most studies addressing these relationships are likely to be biased due to the inclusion of only severe COVID-19 cases admitted to the hospital. Whether the relationships of frailty and multimorbidity with COVID-19 infection and outcomes hold for non-severe forms of COVID-19, it is of major interest to improve risk stratification and public health guidance in the general population.[27]

Genetic and biomedical factors are not the only determinants that may influence COVID-19 outcomes. During previous pandemics, lower socioeconomic status was associated with a higher disease burden. It is therefore important to consider social vulnerability.[28] Several studies suggest that COVID-19, like other infectious diseases, disproportionally affects socially disadvantaged population groups and places them at a higher risk of disease severity. People of low socioeconomic status, ethnic minority groups[29] and crowded households[30] had higher risks of acquiring COVID-19 and higher hospitalisation rates. Overall, a large number of studies with very different methodological approaches (settings and type of adjustment) have been conducted to assess the role of socioeconomic factors in SARS-CoV-2 infection and outcomes. Therefore, it is important to synthesise the current knowledge from well-designed population-based studies in order to clarify and quantify the weight of each socioeconomic factor.

Finally, most of the investigated COVID-19 related health impacts appear so far to be of short term, for example, hospitalisations, ICU admissions or mortality. Evidence on long-term effects such as potential mental health difficulties, impact on quality of life or functioning, among others, in COVID-19 patients is limited,[31 32] but it will certainly grow as the pandemic prolongs.

The previous assumptions justify the relevance, need and importance of summarising current knowledge using a systematic review approach regarding the aetiological and prognostic roles of frailty, multimorbidity and socioeconomic characteristics in developing COVID-19 disease and related short-term and long-term health consequences, among general population. The PROSPERO registry for systematic reviews[33] was cross-checked to ensure the novelty of the research questions.

## Objectives

This paper aims to describe the methodology, which will be used for conducting systematic reviews intended at synthesising knowledge on the roles of frailty, multimorbidity and socioeconomic characteristics in the acquisition and evolution of COVID-19 related health outcomes. Two main study objectives have been determined.

The first objective is to explore the effects of frailty, multimorbidity and socioeconomic characteristics on SARS-CoV-2 infection risk in the general population. The goal is to consider the general population at risk of developing SARS-CoV-2 infection or related outcomes and to estimate a potential increase in risk associated with a given risk factor (frailty or multimorbidity or socioeconomic characteristics). The outcomes of interest are infection by SARS-CoV-2, hospitalisation for COVID-19, ICU admission, mechanical ventilation and mortality due to COVID-19.

The second objective is to explore the prognostic effects of frailty, multimorbidity and socioeconomic characteristics in SARS-CoV-2 infected subjects from the general population. The goal is to consider SARS-CoV-2 infected individuals from the general population and estimate a potential increase in risk of developing severe short-term and long-term outcomes associated with a given risk factor (frailty, multimorbidity or socioeconomic characteristics). The outcomes of interest are death due to COVID-19, hospitalisation, ICU admission, mechanical ventilation, functioning, quality of life, disability, mental health difficulties and work absence/sick leave.

## METHODS AND ANALYSIS
## Protocol and registration

This protocol is written along the requirements of Preferred Reporting Items for Systematic Review and Meta-Analysis Protocols 2015 checklist.[34] The study

is registered in the PROSPERO registry for systematic review protocols under the registration number: CRD42021249444.

Should there be any substantial amendments to this protocol in the course of the study, they will be fully documented and included in dissemination, such as reports and scientific publications.

## Eligibility criteria

We applied the Population, Exposure, Comparator and Outcomes framework[35] to clearly specify research objectives. This framework has been increasingly recognised in population health, environmental and occupational health fields due to the importance of identifying the exposure and clarifying its association with the outcome using comparator groups.[35 36] The study objectives are hence further detailed as follows:

### Objective 1

Population (P)=general or well-defined population SARS-CoV-2 negative.

Exposure (E)=(either) frailty, multimorbidity and lower socioeconomic status.

Comparator (C)=general population with no frailty or no multimorbidity or with better socioeconomic status.

Outcome(s) (O)=infection by SARS-COV-2, hospitalisation for COVID-19, ICU admission, mechanical ventilation and mortality due to COVID-19.

### Objective 2

Population (P)=general or well-defined population SARS-CoV-2 infected (COVID-19 diagnosed, with for example, PCR test, medical imaging or similar).

Exposure (E)=(either) frailty, multimorbidity and lower socioeconomic status.

Comparator (C)=SARS-CoV-2 infected subjects from the population with no frailty or no multimorbidity or with better socioeconomic status.

Outcome(s) (O)=death due to COVID-19, hospitalisation for COVID-19, ICU admission, mechanical ventilation, functioning, quality of life, disability, mental health difficulties and work absence/sick leave.

A widely agreed definition of multimorbidity is lacking in the scientific literature[37 38]; however, the most frequently cited definition is a coexistence of two or more conditions within an individual,[12 38] which will likewise apply in this review. Multimorbidity can also be operationalised differently, in a sense that it is commonly explored as a disease count (≥2, ≥3 diseases or similar), through comorbidity/multimorbidity indices or disease combinations. To summarise all relevant evidence, studies with any of the valid mentioned measurement tools that observe the association between minimum of two conditions and COVID-19 related outcomes will be retained. In the final summary table, this review will specify the multimorbidity measurement tool and the number and list of conditions for each included study.

Similarly, number of definitions for frailty can be found in the literature where each holds its advantages and limitations. Two main approaches coexist and are widely used to define frailty: the phenotypic approach described by Fried *et al*[13] in 2001 and the deficit accumulation approach proposed by Rockwood *et al*[14] in 2004. We choose to include all studies using a definition of frailty based on one of these two approaches, provided that its predictive value had been previously validated. Studies using the term 'frailty' without reference to either of these two standards will not be included.

Likewise, literature looking at the role of the socioeconomic gradient in COVID-19 related outcomes will be retained. Socioeconomic status assessment will include standard indicators, such as income, education, occupation, employment, housing, urban/rural setting, household size, race and marital status measured at the individual or the community level (ecological studies). All age groups will be taken into account considering that socioeconomic differences may be observed across all ages and that multimorbidity is increasingly present among younger individuals as well as among older adults.[39]

## Eligible studies and settings

To identify population based studies, the following definitions will apply: 'a population-based study is defined as a study of properties of a well-defined population, such as individuals residing in a defined geographic region in a given time period'[40] and 'population-based studies aim to answer research questions for defined populations; answers should be generalizable to the whole population addressed in the study hypothesis, not only to the individuals included in the study'.[41] Therefore, the following examples of population-based studies may be eligible for our reviews: community-dwelling, hospital-based, nursing homes, homes for elderly or similar, as long as the population in the study is representative of the country or region from which the sample was drawn. Only quantitative studies will be considered.

## Study designs

All types of aetiological and prognostic observational studies with comparator groups will be eligible, such as cohorts, cross-sectional studies, case–control studies or ecological studies (for socioeconomic determinants only). Interventions and clinical trials will be excluded, as well as qualitative and case studies. Only published peer-reviewed original studies written in English will be considered.

## Information sources and search strategy

The following databases were explored: PubMed, Embase, WHO COVID-19 Global literature on coronavirus disease[42] and PsycINFO. The search strategy considers all possible variations of the terms: multimorbidity, frailty, terms defining socioeconomic characteristics, COVID-19 and study design types. The search strategy was adjusted to the technical specificities of each database. The

**Table 1** PubMed search strategy

| | | |
|---|---|---|
| #1 | Covid-19 (adjusted from Lazarus et al)[64] | (((((((((((((((((("Betacoronavirus"[MeSH Terms] OR "Coronavirus Infections"[MeSH Terms]) OR "COVID-19"[Supplementary Concept]) OR "Coronavirus"[MeSH Terms]) OR "Severe Acute Respiratory Syndrome Coronavirus 2"[Supplementary Concept]) OR "2019nCoV"[All Fields]) OR "betacoronavirus*"[All Fields]) OR "corona virus*"[All Fields]) OR "coronavirus*"[All Fields]) OR "coronovirus*"[All Fields]) OR "CoV"[All Fields]) OR "CoV2"[All Fields]) OR "COVID"[All Fields]) OR (("COVID-19"[Supplementary Concept] OR "COVID-19"[All Fields]) OR "covid19"[All Fields])) OR (((((((("COVID-19"[All Fields] OR "covid 2019"[All Fields]) OR "Severe Acute Respiratory Syndrome Coronavirus 2"[Supplementary Concept]) OR "Severe Acute Respiratory Syndrome Coronavirus 2"[All Fields]) OR "2019 ncov"[All Fields]) OR "SARS CoV 2"[All Fields]) OR "2019nCoV"[All Fields]) OR (("wuhan"[All Fields] AND ("Coronavirus"[MeSH Terms] OR "Coronavirus"[All Fields])) AND (2019/12/1:2019/12/31[Date - Publication] OR 2020/1/1:2020/12/31[Date - Publication])))) OR "HCoV-19"[All Fields]) OR "nCoV"[All Fields]) OR "SARS CoV 2"[All Fields]) OR "SARS2"[All Fields]) OR "SARSCoV"[All Fields]) OR (((("sars virus"[MeSH Terms] OR ("sars"[All Fields] AND "virus"[All Fields])) OR "sars virus"[All Fields]) OR ("sars"[All Fields] AND "CoV"[All Fields])) OR "sars cov"[All Fields])) OR (("Severe Acute Respiratory Syndrome Coronavirus 2"[Supplementary Concept] OR "Severe Acute Respiratory Syndrome Coronavirus 2"[All Fields]) OR "SARS CoV 2"[All Fields])) OR "severe acute respiratory syndrome cov*"[All Fields]) AND (2019/11/17:3000/12/31[Date - Entry] OR 2019/11/17:3000/12/31[Date - Publication]) OR "COVID-19"[MeSH Terms] OR "SARS-Cov-2"[MeSH Terms] <br> OR "SARS CoV-2" OR "SARS-CoV-2" OR SARSCoV2 OR "CoV-2" OR "covid 19" OR covid2019 OR "covid-2019" OR "novel CoV" OR "corona pandemic*" OR "wuhan virus*" OR "CoV 2" OR ((wuhan OR hubei OR huanan) AND ("severe acute respiratory" OR pneumonia*) AND outbreak*) |
| #2 | Frailty | frailty OR frail OR frailty[MeSH Terms] |
| #3 | Multimorbidity (adjusted from Makovski et al)[65] | multimorbidity OR "multi-morbidity" OR "multi morbidity" OR multimorbidities OR "multi-morbidities" OR "multi morbidities" OR multimorbid OR "multi-morbid" OR "multi morbid" OR comorbidity OR "co-morbidity" OR "co morbidity" OR comorbidities OR "co-morbidities" OR "co morbidities" OR comorbid OR "co morbid" OR "multiple chronic conditions" OR "multiple chronic diseases" OR "multiple conditions" OR "multiple diseases" OR "multiple disorders" OR polymorbid* OR "poly-morbid*" OR "poly morbid*" OR polypath* OR pluripath* OR multipath* OR "multi path*" OR "multi-path*" OR "multiple pathologies" OR "disease cluster" OR "disease clusters" OR "disease pattern" OR "disease patterns" OR "concurrent chronic diseases" OR "multiple chronic disorders" OR multimorbidity[MeSH Terms] OR comorbidity[MeSH Terms] |
| #4 | Socioeconomic characteristics | "socio economic" OR "socio-economic" OR socioeconomics OR "socio-economics" OR "socio economics" OR socioeconomic OR "social difference" OR "social differences" OR "social inequality" OR "social inequalities" OR "socioeconomic inequality" OR "socioeconomic inequalities" OR "social disparity" OR "social disparities" OR education OR literacy OR "socioprofessional" OR "socio-professional" OR "socio professional" OR "social conditions" OR "social class" OR "social classes" OR "social class"[MeSH Terms] OR "socioeconomic factors"[MeSH Terms] OR health status disparities[MeSH Terms] OR income OR poverty OR deprivation OR rural OR urban OR "housing deprivation" OR homeless OR houseless OR homelessness OR ethnic* OR race OR emigrant OR immigrant OR migrant OR "minority group" OR "minority groups" OR disadvantaged OR "marital status" OR ((characteristics OR factors OR status) AND (economic OR social OR educational)) OR ((composition OR characteristics OR size) AND (family OR household)) |
| #5 | Study design | "cross-sectional" OR "cross sectional" OR "case-control" OR "case control" OR cohort OR longitudinal OR "ecological study" OR "ecological studies" OR "ecological design" OR "ecological designs" OR observational OR "observational study" OR "observational studies" OR "observational design" OR "observational designs" OR "prospective study" OR "prospective studies" OR "prospective design" OR "prospective designs" OR "retrospective study" OR "retrospective studies" OR "retrospective design" OR "retrospective designs" OR "prospective observational study" OR "prospective observational studies" OR "retrospective observational study" OR "retrospective observational studies" OR case-control studies[MeSH Terms] OR cohort studies[MeSH Terms] OR cross-sectional studies[MeSH Terms] |
| #6 | | #2 OR #3 OR #4 |
| #7 | | #1 AND #5 AND #6 |

example of the search strategy in PubMed is presented in table 1. The search strategy for remaining three databases is presented in online supplemental annex 1.

The search was restricted using date (January 2020–7 April 2021) and language (English) filters. Reference lists of retained studies will be explored for potentially

omitted relevant records. For studies where pertinent data are not presented but seem to be available or if measurements of interest (SARS-CoV-2 diagnosis, outcomes and risk factors) are not clearly defined, study authors will be contacted for details. If deemed relevant, update of the literature search will be performed.

## Study records
### Data management
The literature will be managed using Rayyan,[43] EndNote and Excel. Records retrieved from the four databases will be exported in EndNote for deduplication. Title and abstract screening will be performed in Rayyan, a web and mobile app for systematic reviews. Full-text articles will be read in EndNote, and data will be extracted in customised Excel tables. Two researchers will perform the screening process and data extraction. Considering the actuality of the topic, the quantity of the literature envisaged and the need for timely information, several researchers will be involved in the screening process. For example, two researchers will assume the role of the first reviewer while several colleagues will act as the second reviewer. Weekly meetings will ensure a clear understanding of the study process. In addition, prior to the start of the study selection, a pilot test is foreseen; for instance, homogeneity in the understanding of the exclusion and inclusion criteria among reviewers will be compared and explained on a minimum of 50 studies.

### Selection process
Applying the exclusion criteria displayed in list 1, two reviewers will independently screen all records by title and abstract (records screening phase) to select studies for full-text reading (reports screening phase).[44]

#### List 1: reasons for exclusion from the systematic review (title/abstract or records screening phase)
1=Language other than English.
 2=Not an original research (eg, editorial, protocol, etc) or no original results.
 3=Unrelated topic.
 4=Not a population-based study.
 5=Subpopulation (medical staff, students, pregnant women, etc).
 6=Duplicate.
The criteria above will be used in a hierarchical order to reduce the discrepancy in selected exclusion criteria between the reviewers, implying that should there be more than one criterion applicable to a study, the one higher on the list will be assigned.

Exclusion criteria in list 2 will apply for selecting studies during a full-text (reports) reading phase.

#### List 2: reasons for exclusion from the systematic review (full-text reading or reports screening phase)
1=Not a population-based study.
 2=SARS-CoV-2 infection diagnosis not clear.*

3=Study does not consider people with frailty or multimorbidity OR does not contain information on socioeconomic characteristics.
 4=Outcome not within the scope of our objectives.
 5=Outcome measurement tool not clear.*
 6=Risk factor measurement tool not clear.*
 7=Subpopulation (eg, pregnant women, healthcare workers, students etc).
 8=Not an original research (eg, editorial, protocol, review, conference abstract, grey literature, etc), no original results or not peer reviewed.
 9=The same or largely the same population already considered in another study for the same outcome.
 10=Clinical trial or intervention study.
 11=Qualitative study.
 12=Descriptive study, absence of a comparator group and/or no measure of the association of interest.
 13=Other (explain).
*Even after contacting authors.

The records will be rated as eligible/not eligible/questionable[45] at both screening stages. Any disagreement between the reviewers will be resolved by consensus in referral to this protocol; a third reviewer will be consulted if necessary. Disagreement between the reviewers will be reported in percentages.[46]

During the screening process, all articles will be sorted by type of exposure factors, for example, frailty and multimorbidity on one side (as biomedical determinants) and socioeconomic characteristics on the other, and by objective (objective 1 or 2). Articles on biomedical determinants will later be divided to studies on frailty or multimorbidity.

Retained studies will be kept for further analyses.

## Data items
Six extraction tables for each exposure and objective will be prepared. They will include: study title, authors, year of publication, study design, study setting and duration, population characteristics, sample size, COVID-19 diagnostic criteria, risk factors and study outcomes and means by which they were assessed, measurement of the association between variables of interest and study adjustment factors.

Two reviewers will extract data independently and compare for agreement.[46]

## Risk of bias in individual studies
Two reviewers will separately assess a risk of bias for each study retained for final reporting. Any disagreement will be resolved by consensus; otherwise, a third reviewer will be consulted. The Newcastle-Ottawa scales will be used to assess the quality of case–control and cohort studies.[47] Each of these scales contains eight items grouped in three categories: selection, comparability and outcome or exposure (for cohort and case–control studies, respectively). Selection items refer to, for example, representativeness of the sample, ascertainment of the exposure, definitions of case and control group, etc. Comparability items

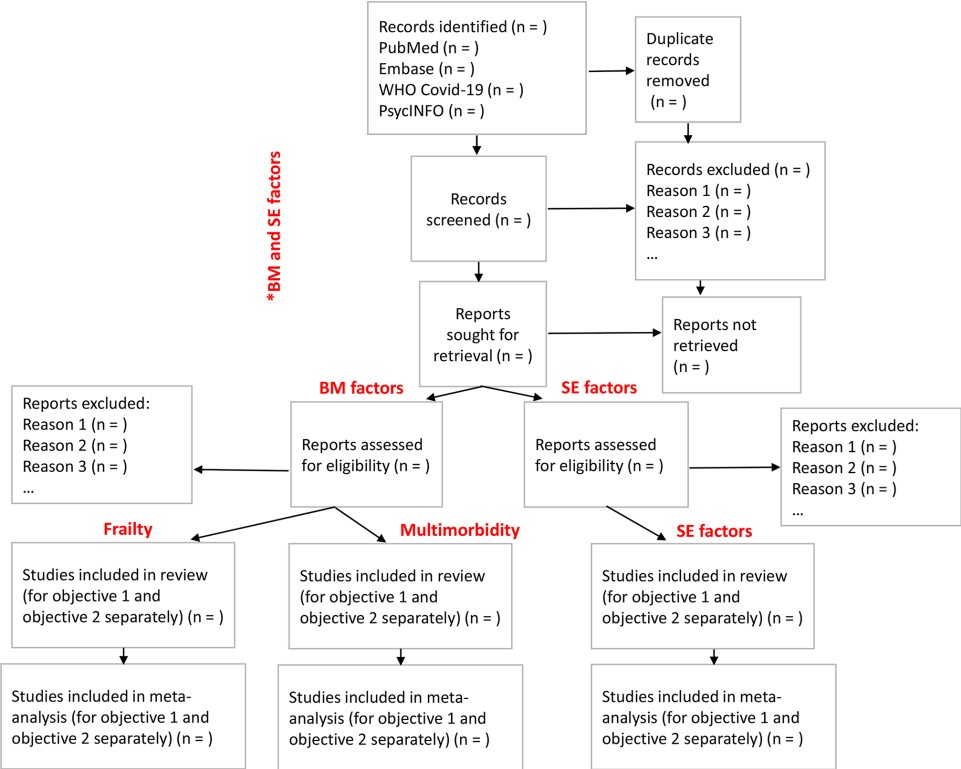

**Figure 1** PRISMA 2020 flow diagram for new systematic reviews, which included searches of databases and registers only (adjusted). *BM, biomedical; PRISMA, Preferred Reporting Items for Systematic Reviews and Meta-Analyses; SE, socioeconomic.

inquire on comparability between, for example, cases and controls or between cohorts based on the study design or analysis. Outcomes category items account for the assessment of the outcome or adequacy of the follow-up, while exposure items account for, for example, ascertainment of the exposure or response rate. Each scale item can carry a maximum of one point, except for the comparability group, where up to two points for an item can be assigned, totalling a maximum of nine points for study quality. Higher score indicates higher study quality.

An adjusted version of the Newcastle-Ottawa scale will be used to assess quality for cross-sectional studies.[48 49] Here maximum of 10 points can be assigned.

Previously mentioned scales will also apply to assess study quality of ecological studies (for socioeconomic risk factor only), depending of the ecological study design.

### Data synthesis

The literature review is ongoing. The search included 10 139 articles in title and abstract screening phase, of which 411 were retained for full-text reading. The screening process results will be presented with Preferred Reporting Items for Systematic Reviews and Meta-Analyses (PRISMA) diagram.[50] Given the interest in several risk factors, the PRISMA diagram may be adjusted, as suggested in figure 1.

The final set of retained studies will be described using standardised tables. They will be summarised within each objective, by type of risk factor and study outcome and presented in tabulated form. Study characteristics will be described as well as their main findings.

When possible, meta-analysis models will be performed by each objective, for each risk factor and for a specific outcome with the aim to better understand the uncertainties (random error and bias) of the obtained results. The feasibility of performing meta-analyses will depend on additional criteria, such as number of studies applying the same measurement tools for risk factors and evaluating the same outcomes, as well as reporting OR, HR, risk ratio or slope. For the meta-analysis, a minimum of four studies will be required. In studies assessing the role of socioeconomic inequalities on health outcomes, unadjusted studies stratified by age group will be sought to avoid overadjustment.[51] For studies assessing the roles of biomedical factors, adjusted analyses will be preferred. Adjustment factors will be thoroughly reported. Random effects meta-analyses will be performed with R software using the metafor package.[52] Findings will be presented with forest plot(s). Publication bias will be visually assessed using funnel plots and formally tested using Begg and Mazumdar's rank correlation test[53] and Egger's regression test.[54] When significant publication bias will be present, Duval and Tweedie's trim-and-fill method will be used to explore the impact on model estimates.[55] Heterogeneity will be explored by computing $I^2$ statistic.[56] When feasible, sensitivity analysis or meta-regression using study quality as a covariate will be conducted. Other covariates

such as proper operationalisation of multimorbidity or frailty (eg, number and list of diseases clearly provided or proper definition used), mean age or gender proportion may also be considered.

## Patient and public involvement

Patients and/or the public were not involved in the design of the protocol and will not be involved in conducting the study, neither in reporting or dissemination plans of the research.

## DISCUSSION

This protocol aims to present the methodology that will be applied to systematically assess the evidence on the aetiological and prognostic roles of multimorbidity, frailty and socioeconomic risk factors in the development of SARS-CoV-2 infection and COVID-19 related severe short-term and long-term health outcomes, among the general population. These reviews should inform on the quantity and quality of the relevant literature and summarise the existing evidence. In addition, the reviews will report which were the preferred assessment methods for the risk factors and most frequent study outcomes considered, as well as which were the statistical approaches and adjustment factors across studies.

To date, no systematic review on the link between multimorbidity and SARS-CoV-2 infection and outcomes has been identified. Even though not specifically focusing on patients with multimorbidity, Tuty Kuswardhani et al[57] have summarised evidence on the association between the Charlson's score and poor short-term outcomes in patients with COVID-19. This score is still frequently used to assess health status in patients with multiple conditions, in the absence of a more specific and widely accepted multimorbidity tool. We plan to elaborate on other multimorbidity measurements and broader COVID-19 related consequences and investigate the possible aetiological role of multimorbidity in the illness acquisition and development. In the context of ageing societies where an increasing number of people live with multiple conditions, scarcity of evidence and public health advice for this population group in the COVID-19 crisis is not acceptable. It is essential to inform how to best evaluate the risks for people with multiple number and combinations of diseases to provide clinicians and policy makers with adequate advice for the most optimal and evidence-based decisions.[27]

On the other hand, several systematic reviews have been conducted among frail population.[22 25] However, these focus on hospital settings, usually single centre or few multicentre settings and short-term outcomes; prognostic roles of frailty are also mainly explored. None of the identified reviews investigated the role of frailty in the development of COVID-19 related outcomes in population representative samples. Our systematic review intends to build on the existing evidence by examining the occurrence of a wide number of COVID-19 health outcomes

accounting for aetiological as well as prognostic roles of frailty, among the general population.

Similarly, reviews show existing disparities in COVID-19 outcomes in lower socioeconomic groups and ethnic minorities. Specific contexts such as low level of education, poor housing conditions and crowded households could play a role in worsening COVID-19 outcomes.[58–60] As previously stated, no systematic review to date has investigated COVID-19 outcomes and severity related to socioeconomic gradient in well-defined population based studies. We, therefore, aim to fill this gap. Whereas the three types of factors (multimorbidity, frailty and socioeconomic factors) will be considered and analysed separately, the similar use of a population-based approach justifies the definition of a common methodology and the sharing of the first steps of the reviews performed.

A first limitation of these systematic reviews is the focus on quantitative studies, using comparator groups and providing measures of associations between considered exposures and outcomes. Case and qualitative studies, which may be informative of more complex connections between multimorbidity, frailty, socioeconomic status and COVID-19, but whose evidence synthesis require the use of methodologies different from systematic reviews of quantitative studies,[61–63] will be left aside for the sake of feasibility. A second limitation of this review more specifically concerns socioeconomic status indicators that are often relational and contextual (to a country, a region, a city) and that heterogeneity may result from their use, precluding meta-analyses in several cases. A third potential limitation of the reviews is that only publications written in English will be considered. With this, we possibly risk to overlook national population representative studies, which may have been in the necessity of the crisis as well as for local purposes, released in local languages exclusively. However, a publication in peer-reviewed international journals in English may also be regarded as an additional study quality check. A strength and a limitation could be the number of scientists who are planned for this study. Involving several team members may accelerate the review process, which is desired when information is crucial. However, one needs to consider the heterogeneity among reviewers in understanding the study method and the review criteria. We will overcome this by conducting a pilot test and holding regular team and individual meetings with involved parties. Moreover, the two first reviewers, experts in respectful areas, will remain the same throughout the screening and data extraction process. Accurately following validated guidelines in preparing this protocol, conducting and reporting the study will help to limit any possible biases to a minimum.

It is worth to mention a high quantity and speed of which the literature is produced during the COVID-19 crisis, as well as a high pace at which data become obsolete. Our reviews intend to provide some of the first estimates of the association between the chosen risk factors and various COVID-19 related outcomes at the population level, by compiling evidence published in a limited timeframe.

We will critically and prudently appraise the findings by evaluating all selected studies for quality. However, the evidence continues to grow rapidly. We believe that our search strategy could be useful in performing regular updates of the literature and therefore help evolve the current knowledge.

We are also certain that information retrieved in our reviews will be useful in clarifying the risks for certain population groups, as for example chosen here, and in orienting faster the optimal health management and prevention actions for other potential health crises should they arise.

We are hopeful that these reviews will provide relevant and timely novel evidence to facilitate public health decisions in the midst of the COVID-19 crisis and identify current gaps in the literature to direct further research.

## ETHICS AND DISSEMINATION

The study will be based on published evidence and is exempt from the ethical approval. The dissemination of findings will include publication in a peer-reviewed journal related to the field and presentation at a scientific conference. In addition, a Population Health Information Research Infrastructure project report will be written.

**Author affiliations**
[1]Department of non-communicable diseases and injuries, Santé publique France, Saint-Maurice, Île-de-France, France
[2]Institut de recherche santé et société (IRSS), Université catholique de Louvain, Woluwe-Saint-Lambert, Brussels, Belgium
[3]Institute of Health Information and Statistics of the Czech Republic, Prague, Czech Republic
[4]Instituto de Saúde Ambiental, Faculdade de Medicina, Universidade de Lisboa, Lisboa, Portugal
[5]Área Disciplinar Autónoma de Bioestatística, Faculdade de Medicina, Universidade de Lisboa, Lisboa, Portugal
[6]Directorate for Project Management, National Directorate General for Hospitals, Budapest, Hungary
[7]Universidad Nacional de Educación a Distancia, Madrid, Spain
[8]Department of Epidemiology and Public Health, Sciensano, Brussel, Belgium
[9]Department of Public Health, Institute of Tropical Medicine, Antwerp, Belgium
[10]Department of Translational Physiology, Infectiology and Public Health, Ghent University, Merelbeke, Belgium
[11]National Center of Epidemiology, Instituto de Salud Carlos III, REDISSEC and RICAPPS, Madrid, Spain

**Acknowledgements** We wish to thank Population Health Information Research Infrastructure (PHIRI) coordination team for initiating and financing this project. We would like to thank documentation experts from Santé Publique France, Laetitia Haroutunian and Sandra Kerzanet, for their valuable advice on formulating the search strategy.

**Contributors** LC-B, JCos, BD, PN, NS and MJF identified the research questions. All authors were involved in the conceptualisation of the search strategy. TTM and JG will share the role of a first reviewer, each reviewing half of the retrieved articles; the role of a second reviewer will be divided between LC-B, SMB, RH, RF-S, FPB, JCot, MA, BV, PB and PN. Acting first and the second reviewer will independently evaluate retrieved records for eligibility and inclusion in the systematic reviews and they will independently extract relevant data. The third-party (JCos), if necessary, will resolve any disagreement between the two reviewers. TTM, JG, JCos, BD and LC-B formulated the first version of this article. All authors revised and approved the final version of the article.

**Funding** These systematic reviews are part of PHIRI project (available from: https://cordis.europa.eu/project/id/101018317/fr). PHIRI is funded by the European Union's Horizon 2020 research and innovation programme under grant agreement no: 101018317. The protocol was developed in collaboration with all listed institutions. The funding body was not involved in the design of the study; it will not be involved in the collection, analysis and interpretation of data or in writing the manuscript. The content of this publication will represent the views of the authors only and not of the European Commission or any other body of the European Union. The European Commission does not take any responsibility for use that may be made of the information it will contain.

**Competing interests** None declared.

**Patient and public involvement** Patients and/or the public were not involved in the design, or conduct, or reporting, or dissemination plans of this research.

**Patient consent for publication** Not applicable.

**Provenance and peer review** Not commissioned; externally peer reviewed.

**ORCID iDs**
Tatjana T Makovski http://orcid.org/0000-0002-3334-4809
Paulo Nogueira http://orcid.org/0000-0001-8316-5035

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
