## [Reviewer comments · BMJ Open]

ARTICLE DETAILS

TITLE (PROVISIONAL)	Etiologic and prognostic roles of frailty, multimorbidity and socioeconomic characteristics in the development of SARS-CoV-2 health outcomes: protocol for systematic reviews of population-based studies
AUTHORS	Makovski, Tatjana; Ghattas, Jinane; Monnier Besnard, Stephanie; Ambrozova, Monika; Vasinova, Barbora; Feteira-Santos, Rodrigo; Bezzegh, Peter; Ponce Bollmann, Felipe; Cottam, James; Haneef, Romana; Devleeschauwer, Brecht; Speybroeck, Niko; Nogueira, Paulo; Forjaz, Maria João; Coste, Joel; Carcaillon-Bentata, Laure

VERSION 1 – REVIEW

REVIEWER	Ecks , Stefan The University of Edinburgh
REVIEW RETURNED	07-Jul-2022

GENERAL COMMENTS	This study sets out to explore correlations between higher risk of severe course of illness with COVID-19, and pre-existing conditions in terms of multimorbidity, frailty, and lower socioeconomic status. The study will do a systematic review. The authors identified more than 10,000 studies that meet the inclusion criteria. The methodology follows standard protocols for systematic reviews. The available databases are correctly identified. The search criteria make sense. Points of further reflection: 1) the definition of multimorbidity and frailty used in this review is defined as wide as possible, but definitions of what should count as multimorbidity and/or frailty are still not standardized. Could the definitions of multimorbidity and/or frailty be more clearly defined? 2) Socioeconomic status is poorly defined in absolute levels of e.g. household income. Socioeconomic status is essentially relational and contextual. This makes reliance on purely population-based studies problematic. 3) The authors emphasize exclusion of studies that are not population-based. There is, however, a substantial body of work on relations between multimorbidity and socioeconomic status based on clinical case studies and qualitative work. Much of this work has been done in the syndemics framework, which is related but different to multimorbidity. These studies are not yet included in this review. Relations between multimorbidity and socioeconomic status are typically highly complex and difficult to disentangle, especially if the standard single disease paradigm is used. Case studies can provide a much clearer sense of connections between multimorbidity, socioeconomic status, and COVID-29. Such studies should not be excluded from a systematic review.
--

REVIEWER	Pérez-Zepeda, Mario Ulises Geriatric Epidemiology Unit, Research Department at the Instituto Nacional de Geriatria
REVIEW RETURNED	25-Jul-2022

GENERAL COMMENTS	This is an interesting protocol about a systematic review and possibly a meta-analysis on three essential risk factors that could influence older adults outcomes in the face of COVID. I have only some minor suggestions: - The title is too large, you might want to shorten it. - Into limitations I would advise to add the speed at which data is getting obsolete about this crisis, and maybe more emphasis on how this epidemic could be a model of a stressor for older adults, and could be replicated with any other crisis. - At some point in methods authors state that if something is not clear they will not include it. Could it be possible before excluding something to contact the authors for clarification?
---

VERSION 1 – AUTHOR RESPONSE

Reviewer 1.

This study sets out to explore correlations between higher risk of severe course of illness with COVID-19, and pre-existing conditions in terms of multimorbidity, frailty, and lower socioeconomic status. The study will do a systematic review. The authors identified more than 10,000 studies that meet the inclusion criteria. The methodology follows standard protocols for systematic reviews. The available databases are correctly identified. The search criteria make sense.

Points of further reflection:

Comment 1. *The definition of multimorbidity and frailty used in this review is defined as wide as possible, but definitions of what should count as multimorbidity and/or frailty are still not standardized. Could the definitions of multimorbidity and/or frailty be more clearly defined?*

Response: We thank the reviewer for the comment and agree that the definitions used to define multimorbidity and frailty were broad. The reviewer however, seems to touch upon a much broader ongoing discussion in the field, that is a continuing strive for a consensual definition on multimorbidity in terms of how many and which conditions the concept should entail. For our review, we have used the most widely applied definition such as a coexistence of two or more conditions, while multimorbidity was operationalized through disease count, combination of diseases and comorbidity indices, in line with a current literature on multimorbidity research. Led by our previous experiences in the field, we suspected that studies may differ significantly in terms of conditions they will consider. This difference is common and depends on data the researchers have at hand or a selection of conditions the authors decide to explore. Not to omit the relevant studies, we have therefore decided to consider all reports that observe the link between any two or more diseases and COVID-19 related outcomes regardless of multimorbidity operationalization.

To clarify our reasoning also in the manuscript, we have altered the following paragraph: Page 6, line 173-186. "A widely agreed definition of multimorbidity is lacking in the scientific literature (37, 38), however the most frequently cited definition is a coexistence of two or more conditions within an individual (12, 38) which will likewise apply in this review. Multimorbidity can also be operationalised differently, in a sense that it is commonly explored as a disease count (≥ 2 , ≥ 3 diseases or similar), through comorbidity/multimorbidity indices or disease combinations. To summarize all relevant evidence, studies with any of the valid mentioned measurement tools that observe the association between minimum of two conditions and COVID-19 related outcomes will be retained. In the final

summary table, this review will specify the multimorbidity measurement tool, and the number and list of conditions for each included study.

Similarly, number of definitions for frailty can be found in the literature where each holds its advantages and limitations. Two main approaches coexist and are widely used to define frailty: the phenotypic approach described by Fried et al. in 2001 (13) and the deficit accumulation approach proposed by Rockwood et al. in 2004 (14). We choose to include all studies using a definition of frailty based on one of these two approaches, provided that its predictive value had been previously validated. Studies using the term "frailty" without reference to either of these two standards will not be included."

Comment 2. *Socioeconomic status is poorly defined in absolute levels of e.g. household income. Socioeconomic status is essentially relational and contextual. This makes reliance on purely population-based studies problematic.*

Response: We have detailed the standard indicators of socioeconomic status which will be used in the Method section, Page 6, lines 188-190: "Socioeconomic status assessment will include standard indicators, such as income, education, occupation, employment, housing, urban/rural setting, household size, race and marital status measured at the individual or the community level (ecological studies)."

We agree that these indicators are relational and contextual (to a country, a region, a city) and that heterogeneity may result from their use. We have indicated this limitation in the Discussion section Page 13, lines 364-370: "A first limitation of these systematic reviews is the focus on quantitative studies, using comparator groups and providing measures of associations between considered exposures and outcomes. Case and qualitative studies, which may be informative of more complex connections between multimorbidity, frailty, socioeconomic status, and COVID-19, but whose evidence synthesis require the use of methodologies different from systematic reviews of quantitative studies (63-65), will be left aside for the sake of feasibility. A second limitation of this review more specifically concerns socioeconomic status indicators which are often relational and contextual (to a country, a region, a city) and that heterogeneity may result from their use, precluding meta-analyses in several cases."

Comment 3. *The authors emphasize exclusion of studies that are not population-based. There is, however, a substantial body of work on relations between multimorbidity and socioeconomic status based on clinical case studies and qualitative work. Much of this work has been done in the syndemics framework, which is related but different to multimorbidity. These studies are not yet included in this review. Relations between multimorbidity and socioeconomic status are typically highly complex and difficult to disentangle, especially if the standard single disease paradigm is used. Case studies can provide a much clearer sense of connections between multimorbidity, socioeconomic status, and COVID-19. Such studies should not be excluded from a systematic review.*

Response: We agree with the author's comment and acknowledge that the relation between multimorbidity and socioeconomic status is complex, and may be informed by case and qualitative studies. However, evidence synthesis from such studies require the use of methodologies different from systematic reviews of quantitative studies, and for the sake of feasibility (timeframe, expertise in teams involved) had to be left aside. We have indicated this limitation in the Discussion section, and mentioned appropriate references. Page 13, lines 364-370, as in comment above.

Reviewer 2.

This is an interesting protocol about a systematic review and possibly a meta-analysis on three essential risk factors that could influence older adults outcomes in the face of COVID. I have only some minor suggestions:

Comment 1. *The title is too large, you might want to shorten it.*

Response: Following the reviewer’s comment we have shorten the title, keeping in mind the Editor’s advice to retain the research question, study design and setting in the title: “Etiologic and prognostic roles of frailty, multimorbidity and socioeconomic characteristics in the development of SARS-CoV-2 health outcomes: protocol for systematic reviews of population-based studies”

Comment 2. *Into limitations I would advise to add the speed at which data is getting obsolete about this crisis, and maybe more emphasis on how this epidemic could be a model of a stressor for older adults, and could be replicated with any other crisis.*

Response: We appreciate the reviewer’s comment. We have added the following paragraph in the limitation section: Page 13, lines 381-391: “It is worth to mention a high quantity and speed of which the literature is produced during the COVID-19 crisis, as well as a high pace at which data become obsolete. Our reviews intend to provide some of the first estimates of the association between the chosen risk factors and various COVID-19 related outcomes at the population level, by compiling evidence published in a limited timeframe. We will critically and prudently appraise the findings by evaluating all selected studies for quality. However, the evidence continues to grow rapidly. We believe that our search strategy could be useful in performing regular updates of the literature and therefore help evolve the current knowledge.

We are also certain that information retrieved in our reviews will be useful in clarifying the risks for certain population groups, as for example chosen here, and in orienting faster the optimal health management and prevention actions for other potential health crisis should they arise.”

Comment 3. *At some point in methods authors state that if something is not clear they will not include it. Could it be possible before excluding something to contact the authors for clarification?*

Response: We agree with the reviewer’s suggestion. The authors of individual studies will be contacted when the information was not clear or missing to maximize withholding of the relevant evidence. Accordingly, we have modified the text in the Method section, Pages 8 and 9 lines 217-219: “For studies where pertinent data are not presented but seem to be available or if measurements of interest (SARS-CoV-2 diagnosis, outcomes and risk factors) are not clearly defined, study authors will be contacted for details.” Likewise, we have added an asterisk with an explanation for the concerned exclusion criteria in the List 2 on Page 9 and 10 of the manuscript, lines 254, 258, 259, 268.

VERSION 2 – REVIEW

REVIEWER	Ecks , Stefan The University of Edinburgh
REVIEW RETURNED	08-Sep-2022
GENERAL COMMENTS	Thank you for addressing my suggestions for revision in detail. I still think that it would be possible to go beyond quantitative data to include qualitative studies in a review such as this, though I appreciate that this is more challenging methodologically.